# Informed Consent and Protection of Personal Data in Genetic Research on COVID-19

**DOI:** 10.3390/healthcare10020349

**Published:** 2022-02-11

**Authors:** Piergiorgio Fedeli, Roberto Scendoni, Mariano Cingolani, Marcelo Corrales Compagnucci, Roberto Cirocchi, Nunzia Cannovo

**Affiliations:** 1Shool of Law, University of Camerino, 62032 Camerino, Italy; piergiorgio.fedeli@unicam.it; 2Department of Law, University of Macerata, 62100 Macerata, Italy; r.scendoni@unimc.it; 3Centre for Advanced Studies on Biomedical Innovation Law (CeBIL), Faculty of Law, University of Copenhagen, Karen Blixens Plads 16, DK-2300 Copenhagen, Denmark; marcelo.c.compagnucci@jur.ku.dk; 4Department of Surgical and Biomedical Sciences, University of Perugia, 06132 Perugia, Italy; roberto.cirocchi@unipg.it; 5Ethic Committee, University of Naples, 80138 Napoli, Italy; nunzia.cannovo@gmail.com

**Keywords:** biobanks, genetic factors, COVID-19 infection, phenotypic expression, general data protection, regulation informed consent

## Abstract

The particular characteristics of COVID-19 demand the careful biomedical study of samples from patients who have shown different symptomatology, in order to understand the genetic foundations of its phenotypic expression. Research on genetic material from COVID-19 patients is indispensable for understanding the biological bases for its varied clinical manifestations. The issue of “informed consent” constitutes the crux of the problem in regulating research biobanks, because it concerns the relationship between the person and the parts separated from the body. There are several consensus models that can be adopted, varying from quite restricted models of specific informed consent to forms that allow very broad authorization (open consent). Our current understanding of COVID-19 is incomplete. Thus, we cannot plan, with precision, the research to be conducted on biological samples that have been, or will be, collected from patients infected by the novel coronavirus. Therefore, we suggest utilizing the “participation pact” between researchers and donors, based on a new form of participation in research, which offers a choice based on the principles of solidarity and reciprocity, which represent the communication of “values”. In the last part of this paper, the general data protection regulation concerning the matter is discussed. The treatment of personal data must be performed with explicit goals, and donors must be provided with a clear, transparent explanation of the methods, goals and time of storage. The data must not be provided to unauthorized subjects. In conclusion, open informed consent forms will be necessary for research on individual patients and on populations.

## 1. Introduction

As the global COVID-19 vaccination campaign continues, science and medicine have yet to find definitive solutions to the COVID-19 social and healthcare emergency [1]. A major difficulty lies in the fact that this infection takes different forms, and those infected may be asymptomatic, experience slight upper respiratory symptoms, or suffer grave and potentially fatal functional compromise.

The current clinical management of COVID-19 consists of infection prevention and control measures and supportive care, including supplemental oxygen and mechanical ventilatory support when indicated. On the basis of the genetic characteristics of the virus [2], the following four drugs, implemented during the SARS-CoV-2 pandemic, have been suggested to be useful: an ACE2-based peptide, remdesivir, 3CLpro-1 and a novel vinyl sulfone protease inhibitor [3]. To date, only remdesivir and several neutralizing antibodies have been approved by the United States Food and Drug Administration for the treatment of COVID-19. However, remdesivir’s clinical effects are controversial, and new antiviral drugs are still urgently needed [4]. 

A number of publications have reported the successes and failures of particular treatment plans, some of which are off-label treatments, or phase I/II or III clinical trials on small numbers of patients [5,6,7,8]. The particular characteristics of COVID-19 demand the careful biomedical study of samples from patients who have shown different symptomatology, in order to understand the genetic foundations of its phenotypic expression [9].

The genome-wide association study (GWAS) has shown that there is a correlation between different loci (e.g., 3p21.31) and the severity of and susceptibility to COVID-19 [10,11,12,13].

Since host-mediated lung inflammation was found, the host’s genetic variants associated with critical illness may represent mechanistic targets for therapeutic development [14,15].

In Italy, the analysis of the genetic factors influencing infection with SARS-CoV-2 and the progression of COVID-19 [16] project (called GEFACOVID) will coordinate a genetic study that proposes new diagnostic and treatment solutions to counter COVID-19. It draws together several entities, including universities (Padova, Brescia, Turin, Trieste, Essex, Essen, Madrid, Munich, Maastricht, and Isfahan), institutions (the Spallanzani National Infective Diseases Institute, the Istituto Superiore di Sanità, INSERM Brest, and SINH Shanghai), foundations (Aviralia and Lorenzini), private companies (ThermoFisher (Waltham, MA, USA), BBraun (Melsungen, Germany), ABBVIE (Lake Bluff, IL, USA), and Alfa-Sigma, DaVinciDigital therapeutics (Novartis), and start-ups (Bioscience Genomics (Milano, Italy), Personal Genomics, Diatheva (Cartoceto, Italy), TOMA-Impact Lab Group (Milano, Italy), GenDx (Utrecht, The Netherlands), and PharmGenetics GmbH (Niederalm, Austria)). 

The objectives of the GEFACOVID project are to examine, in detail, the genetic polymorphisms and pathogenetic mechanisms of the virus, as well as genetic, genomic, metabolomic, epidemiological and clinical data, in order to identify the biomarkers of particular vulnerability to the infection, which increase the risk of potentially lethal complications. The issue should also be raised and monitored for minors, given that the number of pediatric patients affected by COVID 19 is on the rise. Indeed, the fewer pediatric patients at the start of the pandemic did not necessarily mean that children were less likely to become infected [17].

Other local biobanks may be established alongside this infrastructure. In fact, in Milan, a COVID-19 biobank has been established to collect and store the blood and tissue samples of individuals infected with SARS-CoV-2 [18]. 

Biobanks, structures for the collection of human biological material, accessible on the basis of specific criteria for the purpose of research on diagnosis and treatments, exist throughout the world, and have been proven to be indispensable for studies on common and rare diseases. As reported by Fedeli et al. [19], there is no single universal definition of “biobank”, but rather numerous national and international proposals united in the definition used in the scientific literature, which indicate that a biobank is “*a structured collection of human biological material accessible on the basis of specific criteria* [20,21]”, “*in accordance with a code of good usage and correct behavior, and with further directions provided by Ethical Committees and Universities* [22]” and “*in which the information contained in the biological material can be connected to a given person* [23]”, “*for diagnostic, therapeutic and research purposes* [24]”. 

The definition itself points to the fact that the development of biobanks for medical–scientific research is inherently complex, involving issues such as the freedom of self-determination, protection of privacy [25], the sharing of genetic data, the ownership of biological samples [26], and, above all, informed consent.

Since, currently, governments and health systems are primarily focused on implementing measures for the prevention and control of the pandemic, some aspects related to genetic research on COVID-19 (informed consent, collection of biological samples, and protection of personal data), although potentially known, can be applied superficially or evaluated incorrectly.

## 2. Informed Consent in Genetic Research

As already affirmed by other researches [27], the theme of “informed consent” constitutes the true crux of the problem in regulating research biobanks, because it concerns the relationship between the person and the parts separated from the body. There are the following two different phases of informed consent: the first concerns the removal of biological material during diagnostic or treatment procedures, and the second concerns the subject’s manifestation of his/her wishes in a free act of self-determination, regarding acts related to his/her health. 

With the removal of biological material, there is separation between this material and the person’s body.

The new status of the biological sample requires careful consideration regarding the expression of the wishes of the subject from whom the sample was taken, regarding the protection of his/her privacy, since the samples contain information about his/her genetic data.

Examining the literature, Fedeli et al. [28] identified the following three fundamental moments in which it is important to obtain consent: When the subject receives information and chooses to donate his/her biological material to the physician;When the data are shared with other healthcare professionals (when the sample and information are added to the biobank);When the obtained data are made known (disclosure) and further sharing of this information with others is planned (for example, external researchers).

We believe that, in the context of genetic research aimed at analyzing the susceptibility of the host to SARS-CoV-2 infection and the complexity of the clinical expression of COVID-19, it is essential to illustrate to patients that the information obtained, in terms of susceptibility or not, does not make the individual exempt from following the precautionary hygiene rules and possibly being administered a vaccine. On the one hand, the public wants research to advance so that gains can be made for the health of everyone and benefits can be shared, but, at the same time, people are frightened by the risk of possible discriminatory uses of the information acquired; on the other hand, researchers and research institutes call for greater incentives for their work, appealing to the principle of solidarity, and, at the same time, demand the rights to exploit the intellectual property associated with their discoveries. There are also international stakeholders whose excessive protection of information risks slowing, or even paralyzing, the scientific world.

Thus, it is clear that informed consent is fundamental for the collection of biological samples and related data.

## 3. Types of Informed Consent

Currently, there are many informed consent models [29,30], varying from quite restricted models of *specific informed consent* [31] to forms that allow very broad authorization (*open consent* [32]); in between, there is *partially narrow consent*, *multi-layered consent,* non-restricted *broad consent* [33], and *blanket consent,* which allows any research to use the biological samples and related data.

Our current understanding of COVID-19 is partial, and, thus, we cannot plan, with precision, the research to be conducted on biological samples that have been, or will be, collected from patients infected by the novel coronavirus.

Therefore, *specific informed consent*, *partially narrow consent,* or *multi-layered consent* do not meet the needs of the current situation, and, instead, it appears to be more realistic to propose open consent.

However, how can the donor be assured that they will receive complete information when there is no certain information about the pathology?

One possible solution could be a “participation pact” between researchers and donors, developed by The European Institute of Oncology in Milan, based on a new form of participation in research, which offers a choice based on the principles of solidarity and reciprocity, which represent the communication of “values”. In particular, it has been suggested [34] “that: (1) the concept of reciprocity should be broadened to include not only the participant and the researcher, but also society as a whole; (2) the alliance between researcher, participant and society should restore the concept of trust”.

Given that COVID-19 affects entire populations, it will be necessary to collect genetic material and sensitive data at a broad scale for genetic research on entire populations or subgroups [35], and suitable forms for obtaining informed consent must be proposed.

Evidently, the protection of personal data becomes a complex challenge in this context.

## 4. The Role of the European Community Regulation on the Protection of Personal Data in the Pandemic Era

The European Community Regulation on the protection of personal data n. 2016/679 [36] (General Data Protection Regulation, hereinafter “GDPR”) articulates specific rules for processing personal data. The GDPR establishes general prohibition for processing personal data, unless a valid legal basis for the processing exists. One such legal basis, which is often used, is the consent of the data subject. For GDPR objectives, consent should be distinguished from other consent requirements that meet ethical or procedural standards. By and large, the GDPR requires that consent is “given by a clear affirmative act establishing a freely given, specific, informed and unambiguous indication of the data subject’s agreement to the processing of personal data relating to him or her, such as by a written statement, including by electronic means, or an oral statement” (Recital 3 and Art. 4 (11) GDPR).

Nevertheless, obtaining informed consent during the current COVID-19 pandemic is not always feasible, in particular, in cases of the secondary use of data, due to the impossibility or the disproportionate efforts needed to trace individuals. Even in situations where traceability is possible, there will be cases where it would not be recommended, or even legal, to process sensitive data based on consent, due to questions such as whether the consent is freely given and informed [37]. Thus, it is important to base the processing of special categories of personal data on one of the other legal bases, such as public and vital interest (Arts. 6 (1) (d) and 6 (1) (e) GDPR). 

In addition, Art. 9 of the GDPR lists the conditions for processing special categories of data, such as health or social care and public health data (with a basis in law), which is of particular interest in light of the ongoing COVID-19 pandemic. The GDPR expressly allows the processing of personal data for scientific research purposes if it is “necessary for reasons of public interest in the area of public health”. In the midst of the pandemic, the European Data Protection Board (EDPB) published its “Guidelines 03/2020 on the processing of data concerning health for the purpose of scientific research in the context of COVID-19 outbreak”, and supports research and data sharing under the appropriate legal framework. The EDPB guidelines suggest, inter alia, the use of pseudonymization techniques to mitigate the risk of deidentifying or tracking individuals. In deidentification and pseudonymization, it is possible to use a trusted third party to keep the keys safe. Even in these cases, the EDPB recommends that any processing of personal data has to be transparent, with adequate privacy safeguards in place, and must not be shared with third parties without authorization [38]. 

Further issues to take into account are the principles of purpose limitation and data minimization enshrined in Arts. 5 (1) (b) and 5 (1) (c) of the GDPR. In this context, it is not always viable to fully identify the purpose of personal data processing at the time of collection. Recital 33 of the GDPR allows a degree of flexibility, in terms of the specification and granularity of consent required, and certain secondary uses of the data could fall under the category of scientific research purposes (“*It is often not possible to fully identify the purpose of personal data processing for scientific research purposes at the time of data collection. Therefore, data subjects should be allowed to give their consent to certain areas of scientific research when in keeping with recognised ethical standards for scientific research. Data subjects should have the opportunity to give their consent only to certain areas of research or parts of research projects to the extent allowed by the intended purpose*”). Additional measures and safeguards are required in relation to the principles of data minimization and purpose limitation [3].

In Italy, the one real instrument of reference is the Authorization for Treatment of Genetic Data, issued by the Authority for the Protection of Genetic Data [39], the independent authority to whom Italian lawmakers delegate the responsibility for introducing a systematic set of rules on the subject into Italian law. It is the only important and legally binding source on the subject; however, formally, it is only concerned with the treatment of genetic data, not the organic material from which the data were derived, nor is it responsible for the regulation of biobanks in general.

The treatment of personal data must be performed for explicit goals, and donors must be provided with a clear, transparent explanation of the methods, goals and time of storage. The data must not be provided to unauthorized subjects. In the case of genetic research, the Italian data protection authority (“Garante”), on the basis of indications in the regulation, also dictates that if the obtained result has a significant impact on the prevention or treatment of a given pathology in the sphere of the community, the population and local authorities must be informed (Italian Law 27, December 2019, n. 160). 

The genetic data of the population must not be shared or published, except in an aggregated form. 

Especially in the case of COVID-19, the informed consent form must also indicate the possible risks of discrimination or stigmatization of the community involved [40]. An individual with a predisposition to a more severe COVID-19 disease expression should not be exposed to personal or professional limitations; this also applies to subjects with low susceptibility to COVID-19, who could mistakenly consider this status as concrete evidence that they do not need a vaccine, or that it allows them to not adhere to the precautionary hygiene rules [38]. 

## 5. Conclusions

Research on genetic material from COVID-19 patients is indispensable for understanding the biological bases for its varied clinical manifestations. 

The collection of biological samples has already begun, but our incomplete knowledge about the virus, especially its variants, could strongly limit our ability to provide donors with a full understanding of what their consent entails.

The success of the current investigations should be useful for revealing important information that will help cure COVID-19, but also for any upcoming diseases caused by a new coronavirus, whatever they may be. Providing complete information to the patient not only allows the best therapeutic strategies, in the context of a COVID-19 infection, to be adopted, but also provides a tool that can be used for the entire scientific community, by implementing the research and laying the foundations for creating a dataset of genetic susceptibility.

The incomplete knowledge of COVID-19 means that, as progress is achieved, our research goals will have to be adapted, and, thus, old models of specific informed consent will not suffice.

Inevitably, open informed consent forms will be necessary for research on individual patients and on populations.

There can be a high risk of stigmatization, and the donor population should be provided with a generalized informed consent model that explains the value of the research for the progress of scientific knowledge.

## Data Availability

Not applicable.

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
