# Peer review of "Informed Consent and Protection of Personal Data in Genetic Research on COVID-19"

_healthcare, 2022, doi:10.3390/healthcare10020349_

Round 1
Reviewer 1 Report
lines 95 to 102: Consider revising. "parts separeted from the body" or "indissoluble" seem problematic and could benefit from rephrasing.
line 132: it could be useful to refer to, among others,
Jackson SM, Daverio M, Perez SL, Gesualdo F, Tozzi AE. Improving Informed Consent for Novel Vaccine Research in a Pediatric Hospital Setting Using a Blended Research-Design Approach. Front Pediatr. 2021 Jan 12;8:520803. doi: 10.3389/fped.2020.520803. PMID: 33511090; PMCID: PMC7835206. lines 212-213: although not direclty related to genetic data, the connection between data, IC and COVID19 has been highlighted in various works recently -and you might want to consider some of them. See among others: Garasic, M. D. "Informed consent, clinical research, Covid-19 and contact tracing apps: Some neuroethical concerns." BioLaw Journal (2021): 85-95.Author Response
Added

Reviewer 2 Report
The paper discusses the possible legal bases, according to the GDPR, for processing special categories of data in genetic research for COVID-19.
In my opinion, this paper unfortunately does not bring anything new in the overall discussion. The fact that informed consent is difficult to be ensured for research (for any health research actually, not only for COVID-19) is well-known; the authors discuss on this and they suggest that a solution could be the "Participation Pact". However, this term is not being elaborated, whether it is questionable if it is compliant with the GDPR's concept of consent. In the last part of the paper, the authors state that other legal bases - and not consent - may also be applicable; this is true but it is also well-known. Note that the crucial issue is that appropriate safeguards with respect to protection of personal data need to be in place when conducting research, such as data minimisation and pseudonymisation or anonymisation, according to the article 89 of the GPDR - and this is not discussed at all. Even if a law allows for such a processing, the law should have provisions for appropriate safeguards.
The paper has been submitted as a review paper, but it lacks - in my opinion - of reviewing the field; I would expect, from a review paper on the field, to discuss decisions/opinions from competent Data Protection Authorities, relevant guidelines from competent authorities such as the European Data Protection Board etc. It would also of interest to examine whether national laws allow for such a processing and, if yes, under which safeguards.
Therefore, on the basis of the above comments, I would not support acceptance of the paper.
Author Response
Added

Reviewer 3 Report
This work elaborates on the issue of informed consents and protection of personal data for Covid-19 research.
The paper seems to be a report rather than a solid scientific work. The paper is around 5 pages, with the claimed contributions laid out only in 1 page in section 4. The authors do not clarify the contributions of the paper that would justify a publication in the journal. Sections such as related work are missing, and overall the paper is incomplete with a questionable added value.
Author Response
Added

Round 2
Reviewer 3 Report
For a mini-review report (not a scientific paper) this article can be published.